# A Zipf-plot based normalization method for high-throughput RNA-seq data

**Bin Wang** [ID] *¤

Department of Mathematics and Statistics, University of South Alabama, Mobile, AL, United States of America

¤ Current address: Department of Math and Statistics, University of South Alabama, Mobile, AL, United States of America
* bwang@case.edu

## Abstract

Normalization is crucial in RNA-seq data analyses. Due to the existence of excessive zeros and a large number of small measures, it is challenging to find reliable linear rescaling normalization parameters. We propose a Zipf plot based normalization method (ZN) assuming that all gene profiles have similar upper tail behaviors in their expression distributions. The new normalization method uses global information of all genes in the same profile without gene-level expression alteration. It doesn't require the majority of genes to be not differentially expressed (DE), and can be applied to data where the majority of genes are weakly or not expressed. Two normalization schemes are implemented with ZN: a linear rescaling scheme and a non-linear transformation scheme. The linear rescaling scheme can be applied alone or together with the non-linear normalization scheme. The performance of ZN is benchmarked against five popular linear normalization methods for RNA-seq data. Results show that the linear rescaling normalization scheme by itself works well and is robust. The non-linear normalization scheme can further improve the normalization outcomes and is optional if the Zipf plots show parallel patterns.

## Introduction

Errors are inevitable and exist in many analytical platforms, including microarray, qRT-PCR, and next generation sequencing (NGS) technologies. Errors can be introduced in deep NGS data from different sources including sample handling, library preparation, sequencing, and many others. Substitution error is one type of sequencing error. It plays an important role in low-frequency genetic variant detection and can be corrected experimentally and computationally [1]. NGS data often come with measurement errors. For example, gene expressions can be length-biased. Measures of weakly expressed genes are often truncated to zeros due to sequencing depth or detection limits with the current technologies. When done appropriately, normalization can remove the unwanted artificial variations and hence improve the downstreaming data analyses such as differentially-expressed gene detection. Meanwhile, the normalization procedure by itself can introduce new errors to the analyses. Sometimes, it removes the true variations by causes of interest. With the plummet of sequencing costs in recent years,

**Data Availability Statement:** Gene Expression Omnibus (GEO) by NIH NCBI (accession number "GSE19480").

**Funding:** This work was supported by a grant from the Simons Foundation (#422535, Bin Wang).

**Competing interests:** NO authors have competing interests.

massive RNA-seq gene expression data have become publicly available. As a result, it becomes feasible to make good inferences of the weakly expressed genes via bulk RNA-seq data analysis. However, NGS data normalization is challenging and remains a major barrier to pooling data from different sources (by different laboratories, from different tissues, using different sequencing technologies and others).

In general, the normalization methods for gene expression data can be classified into two categories: linear normalization methods and non-linear normalization methods. For linear normalization methods, a profile is rescaled by diving the expressions in the same profile by a positive normalizing parameter so expressions from different profiles are aligned to the same level for direct comparisons. Popular linear normalization methods include but not limited to the following five methods:

- Total count method (TC): TC assumes that all profiles have the same number of reads. This method is simple and easy to implement. However, the total count of a profile can be greatly impacted by the extremely large measures and the performance of TC can be negatively affected by outliers [2, 3].

- Median normalization method (MED): MED assumes that the gene expression distributions of all profiles have the same center and the median expression of a profile is used as its normalizing parameter [4, 5]. MED is easy to implement and works well when the majority of genes are strongly expressed. However, In cancer related genome data analysis, the common median assumption is very likely to fail. In whole genome sequencing, a lot of genes are not expressed and many genes are weakly expressed. When the next generation sequencing technologies are used, the not expressed genes and the genes expressed below the detectable levels (due to the limits of technologies or experimental settings) have zero measures. MED is not applicable for the zero-inflated gene profiles. MED has been implemented within the DESeq Bioconductor package.

- Upper quartile method (UQ): Similar to MED, UQ assumes that the expression distributions of the two profiles to be normalized have the same upper quartile. UQ has been used as a remedy for gene profiles with excessive zeros or weakly expressed genes. However, UQ is not applicable if the proportions of zero counts are higher than 75%. In microarray or RNA-seq gene expression data, it is not uncommon that the vast majority of genes are weakly or not expressed. When the proportion of weakly or not expressed genes is below but close to 75%, the performance of UQ can be poor as the sample upper quartiles can be small and the variation can be dominated by the measurement errors. UQ has been implemented within the edgeR Bioconductor package.

- Trimmed mean of M-values method (TMM): TMM assumes that the majority of genes are not DE across all profiles. It computes the fold changes (ratios of expressions) of genes in a pair of profiles and excludes the extremely large or small fold changes. The mean fold changes of the remaining genes in a profile is computed as its normalization parameter [4, 6, 7]. TMM works well when the number of genes are large and the majority of genes are strongly expressed. However, when the amount of weakly or not expressed genes is large, the fold changes can be greatly impacted by the measurement errors and hence the performance of TMM will be negatively affected. Also, such an approach can be problematic when only a subset of selected genes are profiled, as there is no guarantee that the majority of the genes are not DE. The performance of TMM also depends on the percentage of values trimmed at the two ends. TMM has been implemented within the edgeR Bioconductor package.

- Relative log expression method (RLE): RLE is similar to TMM. It also assumes that the majority of genes are not DE and the normalizing parameters are determined by using the median fold changes instead of the trimmed means [8–10]. RLE is less sensitive to extreme gene expressions than TMM. However, its performance may not be good when there are excessive weakly or not expressed genes in the profiles. RLE has been implemented in the DESeq and DESeq2 Bioconductor packages.

The non-linear normalization methods are usually based on stronger assumptions and are used for special purposes. For example, reads per kilo-base per million mapped reads (RPKM), fragments per kilo-base per million mapped reads (FPKM), and transcripts per million (TPM) methods are commonly used to correct the length biases for genes of different sizes. The normalization steps of these three methods are similar to TC except an extra adjustment step to correct the biases under an assumption that the expression measures are proportional to the lengths of the genes. The quantile normalization (QN) is another widely used non-linear method which assumes that all profiles follow similar distributions and allows gene-level expression modifications so that all profiles have the same distribution after normalization [11–14].

In this study, we propose a novel normalization method designated for RNA-seq data with excessive zeros and a large number of small counts ($\leq 10$). With the new normalization method, we perform linear and/or non-linear normalization so that the normalized profiles have similar upper tail behaviors as revealed in the Zipf plots.

## Materials and methods

### RNA-seq data for lymphoblastoid cell lines

As part of the International HapMap project, a total of 52,580 RNAs were sequenced from lymphoblastoid cell lines (LCLs) derived from 69 Nigerian individuals generated using Illumina Genome Analyzer II (Homo sapiens) in a study by Pickrell and et al [15]. The reads are either 35 or 46 base pairs (bp) and are mapped using MAQ v0.6.8. The datasets are counts based without normalization, and are publicly available to download on Gene Expression Omnibus (GEO) by NIH NCBI (accession number "GSE19480").

Table 1 shows some summaries of the gene expression measures. The smallest percentage of zero measures among the 129 gene expression profiles is 83.2% and the largest is 86.37%.

**Table 1. Frequency distribution of small counts for the LCL RNA-seq data.**

| Count | Mean (%) | Median (%) | Min (%) | Max (%) |
|---|---|---|---|---|
| 0 | 44469 (84.57) | 44289 (84.23) | 43744 (83.2) | 45412 (86.37) |
| 1 | 803 (1.53) | 805 (1.53) | 385 (0.73) | 1165 (2.22) |
| 2 | 426 (0.81) | 419 (0.8) | 294 (0.56) | 596 (1.13) |
| 3 | 292 (0.56) | 288 (0.55) | 236 (0.45) | 395 (0.75) |
| 4 | 225 (0.43) | 223 (0.42) | 182 (0.35) | 288 (0.55) |
| 5 | 185 (0.35) | 183 (0.35) | 139 (0.26) | 240 (0.46) |
| 6 | 158 (0.3) | 158 (0.3) | 117 (0.22) | 210 (0.4) |
| 7 | 139 (0.26) | 139 (0.26) | 102 (0.19) | 182 (0.35) |
| 8 | 123 (0.23) | 124 (0.24) | 92 (0.17) | 159 (0.3) |
| 9 | 111 (0.21) | 110 (0.21) | 79 (0.15) | 146 (0.28) |
| 10 | 104 (0.2) | 102 (0.19) | 79 (0.15) | 135 (0.26) |

The values in the parentheses are the percentages of the corresponding counts.

That says, if a positive quantile needs to be used as the normalizing parameter to linearly rescale the expressions in a profile, the quantile level cannot be lower than 86.37%. Hence MED and UQ won't work for this data without special handling. In addition, the average percentage of counts with measures less than 10 across all profiles is 89.46% with standard deviation 0.92%. If we simply choose a high level quantile, say the quantile $q_\alpha$ with level $\alpha = 0.90$ or lower, as the normalizing parameter, many profiles may have the same quantile values of $q_\alpha$ due to the existence of the large amounts of tied small counts. As a result, either no normalization will be performed, or small adjustments will be applied to correct the artificial variations in these profiles. In addition, when the normalization parameter is small, the measurement errors or other types of errors can play important roles in the estimated normalizing parameters and hence bias the normalized data. On the contrary, if we choose very high level quantiles as normalizing parameters, the variances of the estimated normalization parameters tend to be large.

## The Zipf plot for NGS gene expression data

Plots (a) and (b) in Fig 1 show the empirical cumulative distribution functions (ECDFs) of the raw gene expressions and the log-transformed gene expressions, respectively. In plot (a), the ECDFs of different profiles are hard to discern due to the wide range of the gene expressions and the large amount of small measures (including the excessive zeros). After the logarithm transformation, the ECDFs in plot (b) show similar patterns for all profiles. However, the upper tails of the ECDFs are still hard to discern.

The Zipf plot is a powerful graphical method to visualize the upper tail behaviors of right-skewed distributions. Let $X = \{X_1, X_2, \ldots, X_n\}$ be a profile with $n$ gene expression measures. Also, let $z_1 > z_2 > \ldots > z_m$ be the $m$ distinct values of $X$ and $f_1, f_2, \ldots, f_m$ be the corresponding frequencies. We assume that the actual gene expression level of a gene is no smaller than the observed count and compute the rank of $z_i$ as follows:

$$r_i = Rank(z_i) = \sum_{j=1}^{n} I(X_j \geq z_i) = \sum_{j=1}^{i} f_j. \tag{1}$$

The plot of the logarithms of ranks versus the logarithms of $z_i$'s is the so-called Zipf plot. Plot (c) of Fig 1 shows the Zipf plots of all LCL profiles. Due to the Zipf plots in plot (c) being densely packed, the profiles can hardly be distinguished from each other. However, the few isolated Zipf plots at the lower part demonstrate obvious patterns of parallel curves.

The Zipf plot has been widely used to check whether a distribution follows the power law. For any $x_{min} > 0$, if $X \geq x_{min}$ follows a power law distribution, the density function and distribution function of $X$ are

$$f(x) = \frac{\kappa - 1}{x_{min}^{1-\kappa}} x^{-\kappa} \quad \text{and} \quad F(x) = 1 - \left(\frac{x}{x_{min}}\right)^{1-\kappa}, \tag{2}$$

where $\kappa > 1$ is the parameter of the power law distribution. In the Zipf plot, rank $r_i$ can be well approximated by $r_i \approx n[1 - F(z_i)]$ when $n$ is sufficiently large, which is often the case for high-throughput RNA-seq data. Now we have

$$\log(r_i) \approx \log(n) - (1 - \kappa)\log(x_{min}) + (1 - \kappa)\log(z_i). \tag{3}$$

For a power law distribution, we expect its Zipf plot to demonstrate a straight line pattern with slope $1 - \kappa < 0$. Plot (c) reveals that expressions of the highly expressed genes in the LCL profiles approximately follow power law distributions. A modified version of the Zipf plot is

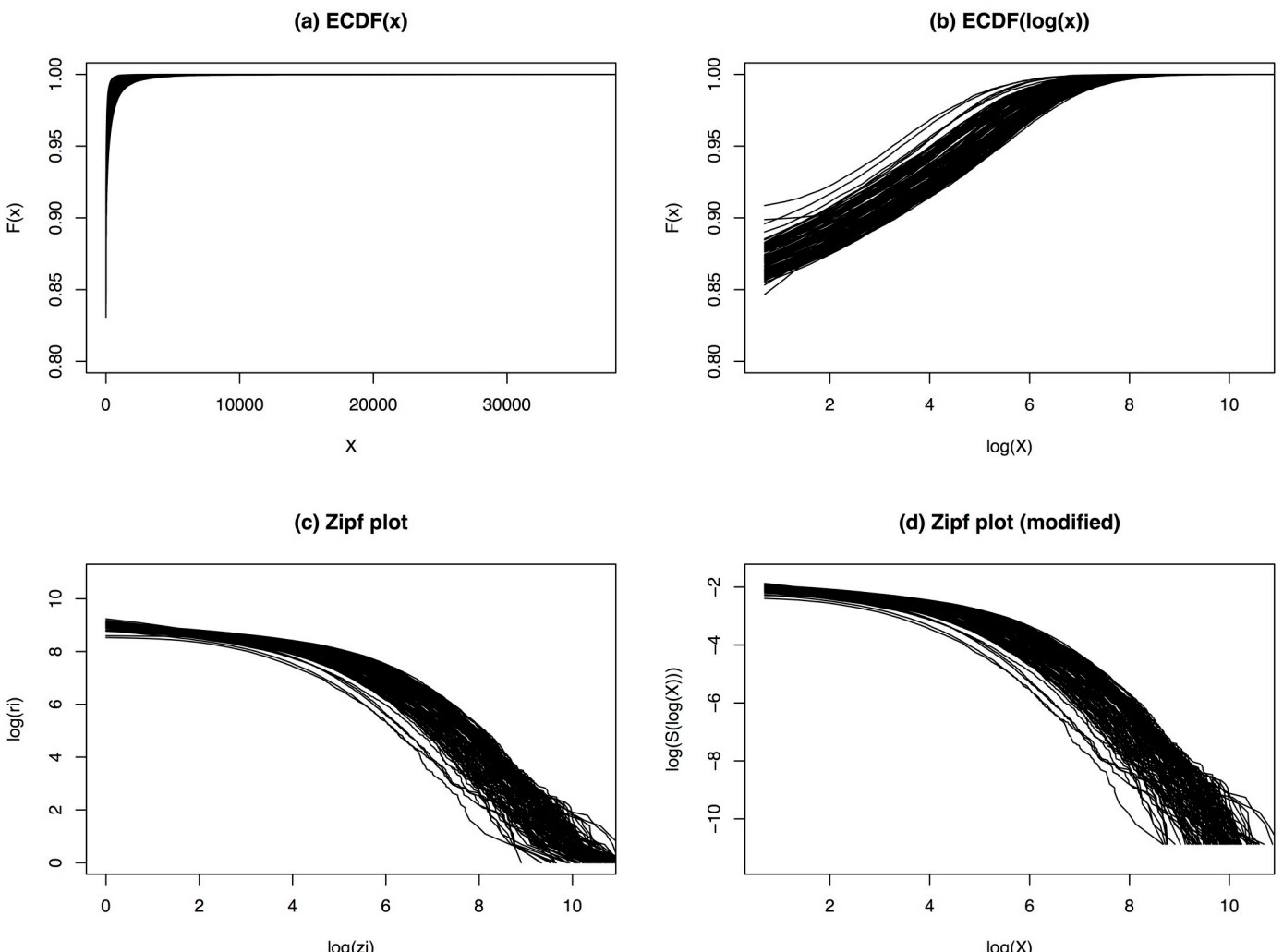

**Fig 1. Similarity comparisons of LCL profiles.** (a) ECDFs of the raw gene expressions; (b) ECDFs of log-transformed gene expressions (zeros not shown); (c) Zipf plots of the gene expressions; (d) Revised Zipf plots by replacing the logarithms of ranks with the logarithms of the approximated survival functions.

displayed in plot (d) by replacing $\log(r_i)$ with $\log(S(z_i)) = \log(1 - F(z_i)) \approx \log(r_i) - \log(n)$, which only results in a downward shift of $\log(n)$ in the y-axis.

## Normalizing NGS gene expression data

For illustration purposes, we choose two profiles whose Zipf plots are far apart. Fig 2 shows the Zipf plots of profile 47 (solid curve) and profile 107 (dashed curve). For convenience, we reuse the notation $X$ and denote a pair of profiles to be normalized as $X$ and $Y$, respectively. Let $z^x = \{z_1^x, z_2^x, \ldots, z_{m_x}^x\}$ be the $m_x$ distinct measures of $X$ and $r^x = \{r_1^x, r_2^x, \ldots, r_{m_x}^x\}$ be the corresponding ranks. Similarly, we denote the distinct measures and ranks of profile $Y$ as $z^y = \{z_1^y, z_2^y, \ldots, z_{m_y}^y\}$ and $r^y = \{r_1^y, r_2^y, \ldots, r_{m_y}^y\}$, respectively. When both profiles follow power law

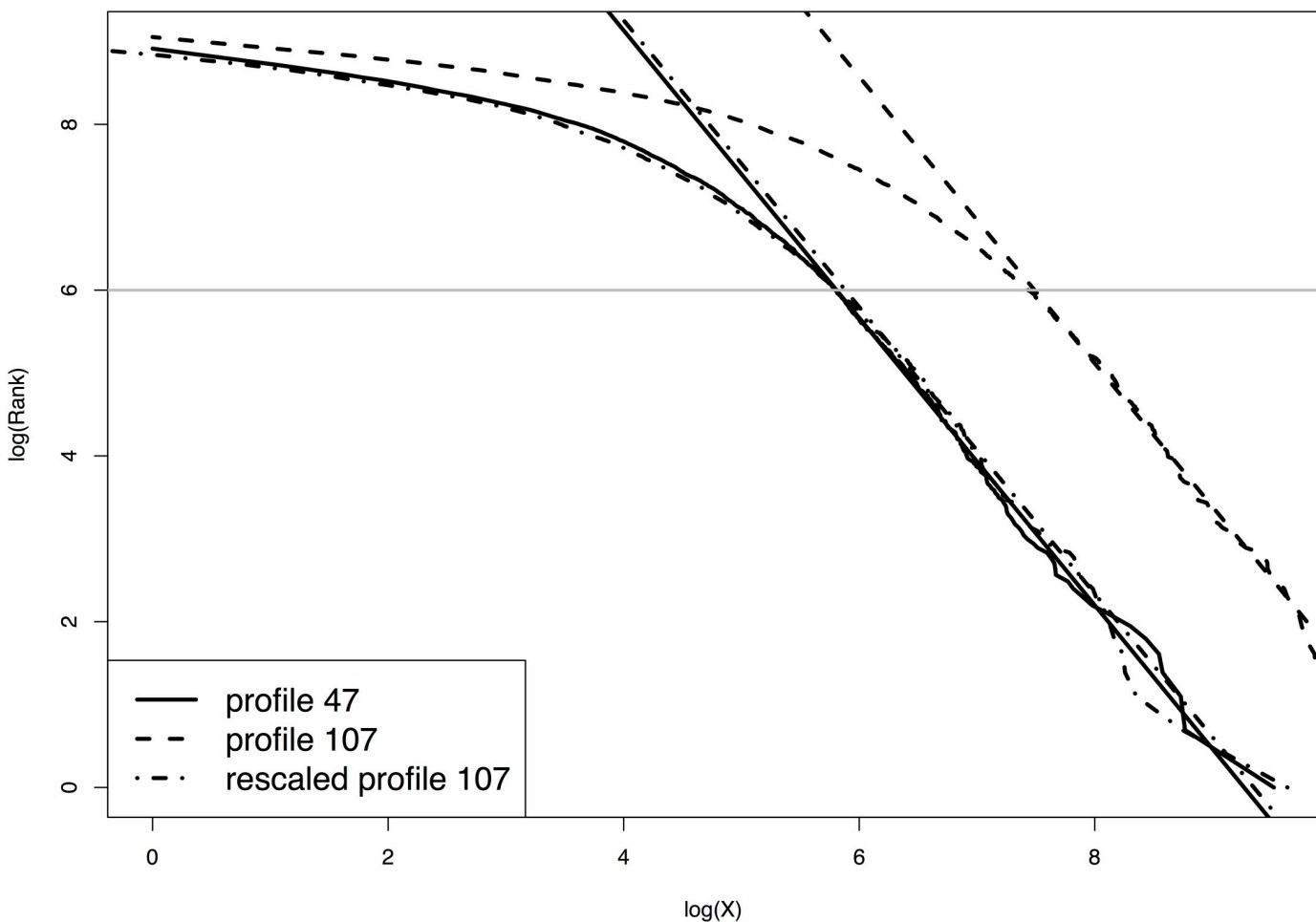

**Fig 2. Normalizing profiles 47 and 107.** Profile 47 has the smallest mean expression (solid curve) and profile 107 has the largest mean expression (dashed curve). The dash-dotted curve shows the Zipf plot of profile 107 after normalization (with both rescaling and power transformation). The solid and dashed straight lines are the fitted least squares lines of log($Rank$) against log($X$) based on genes with log($Rank$)<6 for profiles 47 and 107, respectively. The dash-dotted straight line shows the fitted least squares line for profile 107 after rescaling but before power transformation.

distributions for $X \geq x_{min}$ and $Y \geq y_{min}$, we have

$$\log(r_i^x) = \log(n) - (1 - \kappa_x)\log(x_{min}) + (1 - \kappa_x)\log(z_i^x), \qquad (4)$$

$$\log(r_i^y) = \log(n) - (1 - \kappa_y)\log(y_{min}) + (1 - \kappa_y)\log(z_i^y). \qquad (5)$$

If the highly expressed genes in both profiles follow the same or similar power law distributions, the right ends of the two Zipf plots are expected to show straight line patterns, and the slopes of the fitted least squares lines should be close or the same. In Fig 2, the slope of the fitted least squares line of profile $X$ (solid line) is very close to that of profile $Y$ (dashed line).

Without loss of generality, let $X$ be the reference profile and $Y$ be the profile to be normalized. We normalize profile $Y$ so that the two Zipf plots stay as close as possible after normalization. Graphically, we keep the Zipf plot of profile $X$ unchanged and shift the Zipf plot of profile $Y$ in the x-axis then rotate it, which can be implemented using the following two normalization schemes:

- **Rescaling**: shifting in the x-axis is equivalent to linear rescaling. Let $\sigma$ be the linear normalizing parameter. We have $\log Y' = \log(Y/\sigma) = \log Y - \log \sigma$. Here $\log \sigma$ is the distance that the Zipf plot of $Y$ needs to move towards the Zipf plot of $X$ in the x-axis.

- **Power transformation**: let $\gamma = (1 - \hat{\kappa}_x)/(1 - \hat{\kappa}_y)$ be the non-linear normalizing parameter, which is the ratio of the slopes of the two least squares lines for profiles $X$ and $Y$. We can apply a power transformation $Y' = Y^\gamma$ so the two least squares lines have the same slope.

In summary, profile $Y$ is normalized with respect to reference profile $X$ as

$$Y' = \left(\frac{Y}{\sigma}\right)^{\gamma}, \tag{6}$$

where $\sigma$ and $\gamma$ are the two normalizing parameters. We divide the expressions in the profile to be normalized by $\sigma$ to rescale the expressions so the two profiles will be normalized to close levels. The power transformation with parameter $\gamma$ further improves the similarity between the upper-tail behaviors of the two distributions.

It is worth pointing out that estimating the power law distribution parameters $\kappa_x$ and $\kappa_y$ using the ordinary least squares regression with pre-specified values of $x_{min}$ and $y_{min}$ might not be efficient. We can consider estimating $\kappa_x$, $\kappa_y$, $x_{min}$ and $y_{min}$ altogether based on the highly expressed genes in the two profiles using the Hill estimator [16], or a maximum likelihood estimator as in [17].

As shown in Fig 2, the Zipf plot of profile $Y$ after linear normalization (the dash-dotted curve) is almost identical to the Zipf plot of profile $X$ (solid curve). The slopes of the two fitted least squares lines are very close, which suggests that a non-linear normalization might not be necessary.

## Results

### An example to normalize two gene profiles

Fig 3 shows the MA-plots of profile 47 versus profile 107 before and after normalization in plot (a) and (b), respectively. The MA plot is a commonly used graphical tool to visualize high-throughput sequencing analysis. It first transforms the data in the two profiles onto M (log ratio) and A (average log-expression) scales, then visualizes the differences of the measurements in the two profiles by plotting M against A. In plot (a) we take $M = \log(X) - \log(Y)$ and $A = (\log(X) + \log(Y))/2$, where $X$ and $Y$ are the measures in profiles 47 and 107, respectively. In plot (b), we replace $Y$ with the normalized value $Y'$. The solid curve is fitted using R function `loess` with *span* = 0.2, while the dashed curve is the loess curve with *span* = 0.1. From plot (a), we see that the majority of the points fall below the horizontal line with $M = 0$. This indicates that most genes have larger measures in profile 107 and the expressions need to be scaled down so the two profiles will be brought to close levels. The loess curves show straight line patterns for genes satisfying $A > 5$, which indicates that linear scaling is appropriate for the genes that are not weakly expressed. Meanwhile, plot (a) of Fig 3 shows that the loess curves have different non-linear patterns for points with $A < 2$, which indicates that linear rescaling normalization alone may not be able to achieve good normalization results for weakly expressed genes. In other words, a non-linear transformation is needed to further improve the normalization outcomes.

We choose the highly expressed genes by selecting the expressions with ranks $exp(6) = 403$ or lower and fit least squares lines to these points in the Zipf plots. The slopes of the fitted least squares lines are -1.734 and -1.730 for profiles 47 and 107, respectively. Hence a rough estimate of $\gamma$ is $\hat{\gamma} = (-1.730)/(-1.734) = 0.9977$.

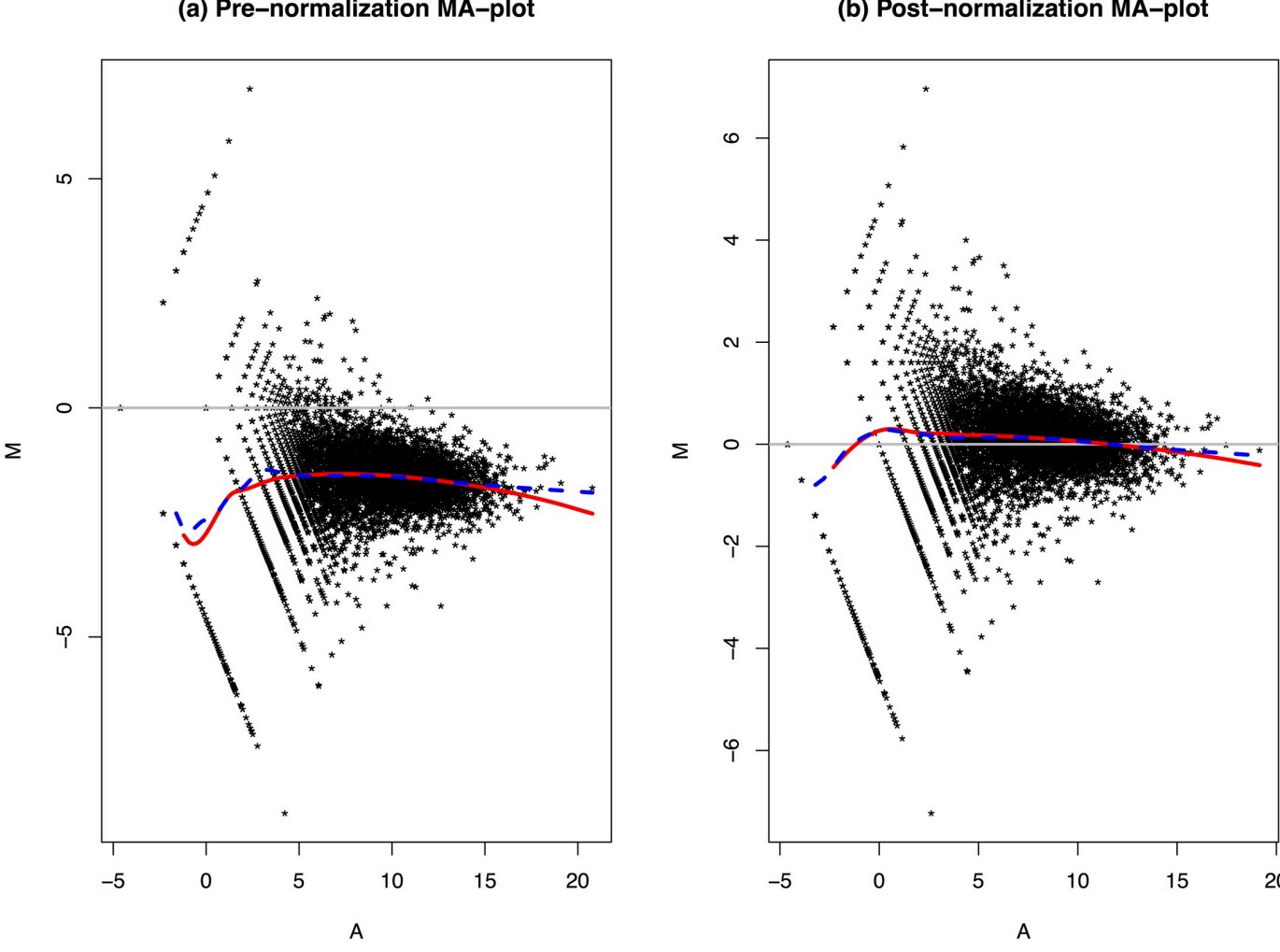

**Fig 3. Comparisons of normalization results for profiles 47 and 107.**

Let $Z = (\log(z_1^x), \log(z_2^x), \ldots, \log(z_1^y), \log(z_2^y), \ldots)$ be a vector of the log-transformed gene expressions of the selected highly expressed genes from the two profiles. Also, let $R = (\log(r_1^x), \log(r_2^x), \ldots, \log(r_1^y), \log(r_2^y), \ldots)$ be a vector of the corresponding log-ranks, and $G = (0, 0, \ldots, 1, 1, \ldots)$ be the group identities. If a point $Z_i$ is from profile $X$, $G_i = 0$, otherwise $G_i = 1$. We then fit a linear model by regressing $Z$ on $R$ and $G$. The estimated coefficient of $G$ is our estimate of $\sigma$.

Plot (b) of Fig 3 shows the MA-plot after normalization. Many genes located above the $M = 0$ line after normalization and the loess curves become more linear than the ones for the pre-normalization data.

## Performance comparison

To evaluate the performance of the proposed normalization method, we normalize all LCL profiles with each of the following methods: TC, MED, UQ, TMM, RLE, and the Zipf plot based normalization (ZN). For fair comparisons, we only use the linear rescaling normalization scheme for ZN (without non-linear normalization). In Fig 1, many Zipf plots stay apart

from each other, which indicates that linear normalization is needed to pull the profiles to similar levels. However, non-linear normalization may not be needed as the Zipf plots show parallel curve patterns in their upper tails. This is also verified by the estimated non-linear normalizing parameter between profiles 47 and 107, which is very close to 1.0 indicating that the power transformation can improve the normalization results but not much.

To reduce the numerical difficulties caused by the zeros, we exclude 39,596 (75.31%) genes with zero across all profiles. A total of 12,984 genes remain in the new dataset, where profile 58 has the smallest number of zeros (20.79%). To improve the performances of MED, UQ, TMM and RLE, we choose profile 58 as the reference profile.

- TC: this method applies to all profiles without any numerical difficulty.

- MED: among all profiles to be normalized, four profiles (3.13%) have zero median. Three of them are normalized using MED after removing all genes with zero counts in both profiles. One profile has zero median after we filter out the genes with zero in both profile. Therefore, all genes with zero in either profile are removed to apply the MED.

- UQ: all profiles have positive upper quartiles in the new dataset. UQ works with no numerical difficulty.

- TMM: to avoid numerical difficulties, genes with zero measures in both profiles are filtered out. We first compute the ratio of expressions in the profile to be normalized and the reference profile for each gene. Then we calculate the M-values by taking the logarithm of the ratios. A total of 15 profiles (11.72%) have infinite means after trimming 30% M-values from both ends. Genes with zeros in either profiles are removed for these 15 profiles to apply TMM.

- RLE: similar to TMM, genes with zero measures in both profiles are removed to compute valid M-values. After that, only one profile doesn't have valid median M-value and genes with zeros in either profile are removed to apply RLE.

- ZN: this method applies to all profiles without any numerical difficulty.

Table 2 shows the summaries of the estimated normalizing parameters for the six selected normalization methods. Results show that MED has overall larger estimated normalizing parameters than the other methods. As a result, only 14.1% of the profiles are upward rescaled with MED while the percentages for the other methods are at least 50%. This is mainly because the reference profile has the smallest number of zeros and the calculation of the median is closely associated with the number of zeros in a profile. In terms of the number of zeros, the selected reference profile has overall strong signals and therefore most of the other profiles are downward rescaled if we use median to compute the normalizing parameter. There are a certain number of profiles not normalized with MED, UQ and RLE. This is because these three methods use a single quantile estimate for each profile to compute the normalizing parameter. Due to the existence of large numbers of small counts in the profiles, tied quantile estimates are likely to be found between a pair of profiles which ultimately results in normalizing parameter estimates of 1.0.

To further benchmark the performances of the six normalization methods, we identify and compare a set of "invariants" based on the normalized data. The idea is as follows: if the majority of genes are not DE and if a normalization method works well, we expect to be able to identify a subset of genes that are strongly expressed with small deviations, namely, the invariants. To take both the deviation and expression level into consideration, we use the coefficient of variation (CV) to determine whether a gene is an invariant. Genes with 50% or more zeros will

**Table 2. Comparisons of estimates of normalizing parameter.**

| Method | Min. | 1st Qu. | Median | Mean | 3rd Qu. | Max. | SD | Scal Up | No Chng |
|--------|------|---------|--------|------|---------|------|-----|---------|---------|
| TC | 0.476 | 0.627 | 0.764 | 0.956 | 1.165 | 3.430 | 0.498 | 0.664 | 0.000 |
| MED | 0.667 | 1.000 | 1.500 | 1.798 | 3.000 | 6.000 | 1.062 | 0.141 | 0.227 |
| UQ | 0.459 | 0.622 | 0.814 | 1.028 | 1.340 | 4.250 | 0.586 | 0.609 | 0.008 |
| TMM | 0.278 | 0.552 | 0.862 | 0.861 | 1.123 | 1.601 | 0.357 | 0.578 | 0.000 |
| RLE | 0.182 | 0.500 | 0.841 | 0.858 | 1.154 | 1.667 | 0.388 | 0.562 | 0.031 |
| ZN | 0.421 | 0.778 | 0.993 | 1.192 | 1.411 | 4.296 | 0.617 | 0.500 | 0.000 |

not be considered as invariants because most of them are weakly expressed and can be significantly impacted by measurement errors. Because the CVs are severely skewed to the right, we perform logarithm transformation to the CVs and show the distributions. In Fig 4, the two curves for TMM (short dash-dotted) and RLE (long dashed) are similar, while the other four methods show similar patterns. The distribution of TC (solid) is very similar to the distribution of UQ (dotted). MED (short dashed) and ZN (long dash-dotted) have similar distributions

**Coefficient of Variation**

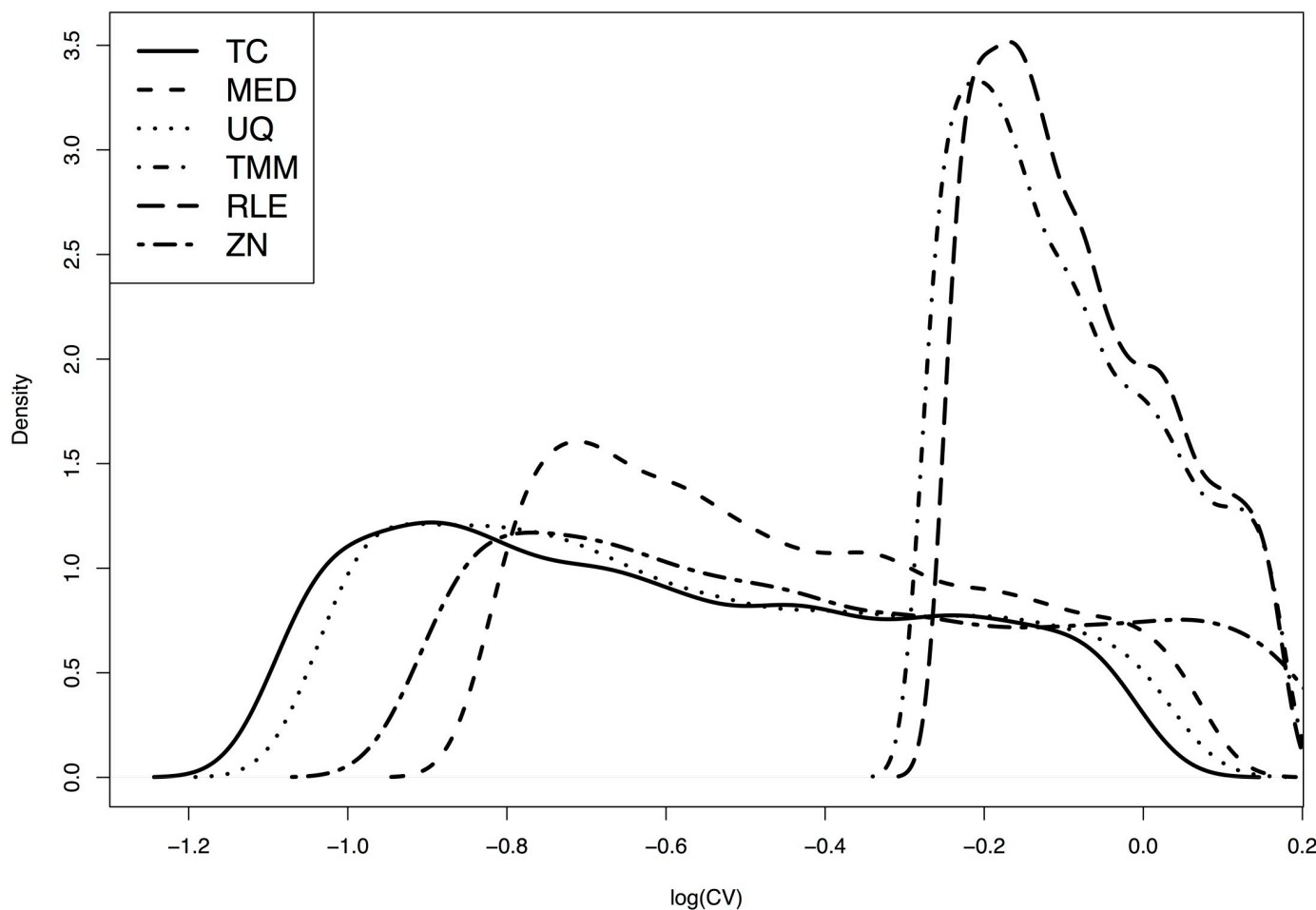

**Fig 4. Distributions of coefficient of variations after logarithm transformation.**

**Table 3. Agreement among different normalization methods.**

| Method | TC | MED | UQ | TMM | RLE | ZN |
|--------|-------|-------|-------|-------|-------|-------|
| TC | **0.298** | 0.917 | 0.981 | 0.892 | 0.879 | 0.987 |
| MED | 490 | **0.402** | 0.955 | 0.739 | 0.711 | 0.918 |
| UQ | 754 | 654 | **0.314** | 0.829 | 0.812 | 0.982 |
| TMM | 470 | 144 | 303 | **0.727** | 0.989 | 0.861 |
| RLE | 455 | 119 | 285 | 911 | **0.749** | 0.846 |
| ZN | 850 | 522 | 787 | 417 | 396 | **0.356** |

and ZN has larger overlaps with TMM and RLE than TC, UQ and MED. It is not surprising that TMM and RLE are close, as both are based on the fold changes and the small counts play important roles in computing the normalizing parameters. TC, UQ and ZN give more weights to the observations to the upper tails and therefore they show similar patterns. If a small cutoff of log-CV is chosen, say -0.4, no gene can be identified as invariants by TMM or RLE.

Table 3 compares the agreement among different normalization methods. The diagonal cells show the cutoff CVs used to identify 1000 invariants with each normalization method. TMM and RLE have much larger cutoff CVs than the other four methods. The numbers in the lower triangular matrix show the number of genes identified by both methods. For example, among the 1000 invariants identified by TC, 850 are also identified by ZN. TMM and RLE have the largest overlap of 911 common invariants. The values in the upper triangular matrix show the Spearman correlation coefficients of the CVs computed based on data normalized using two different methods. The coefficient between TMM and RLE is 0.989, and 0.987 between ZN and TC. The overall agreement among the six normalization methods is good.

We also check the similarity of the upper-tails of all profiles by comparing the slopes of the least square lines fitted on genes with log($Rank$)<6 in their Zipf plots. The 95% confidence interval of the estimated slopes is ($-1.745$, $-1.713$), which confirms that all profiles have similar patterns of upper-tail behaviors and the non-linear normalization can be optional.

## Discussion

The estimates of the two normalization parameters, $\sigma$ and $\gamma$, can be further improved. To find the optimal estimates of ($\sigma, \gamma$) jointly, we search over a fine grid in the neighborhood of the OLS estimates based on the Zipf plots as in Fig 2. We define the optimal estimates as a pair of values of ($\sigma, \gamma$) that minimize the the following objective function:

$$D(\sigma, \gamma) = \max_t |F_n^x(t) - F_n^y(t)|. \tag{7}$$

The rationale that we choose the above Kolmogorov–Smirnov statistic to measure the distance between the distribution functions of the two profiles after normalization is its simple form and the good performance of the empirical distribution function in approximating the true distribution functions for large samples.

Fig 5 shows the perspective plot of the surface of $D$ over the $\sigma$-$\gamma$ plane for profiles 47 and 107. It shows that $D$ doesn't change much as $\gamma$ varies. However, $D$ changes greatly if we fix $\gamma$ and change the value of $\sigma$ alone. Fig 6 shows the MA-plot after normalization using the optimal estimates of ($\sigma, \gamma$) = (4.6261, 0.9874). There are very small differences between the two loess curves fitted to data normalized using the OLS estimates and the optimal estimates, respectively. Due to the lack of ground truth about the gene expressions in different profiles, it is hard to tell which estimate works better.

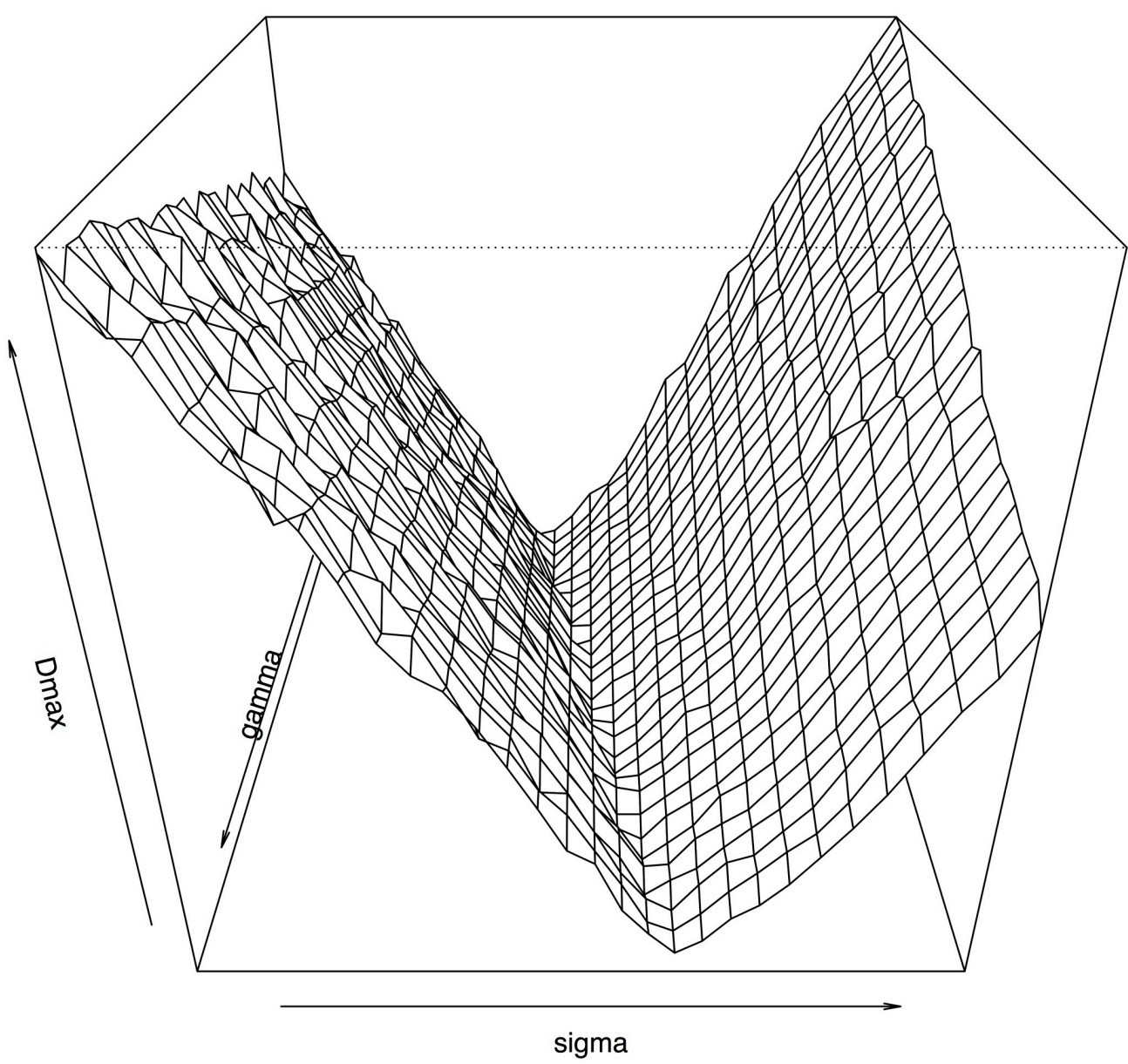

**Fig 5. Optimal estimates of $(\sigma, \gamma)$.**

For genes expressed at very low levels, say $A < 2$, a more aggressive normalization strategy to manipulate the expressions at gene-level is not necessary. The low measures tell us whether the genes are detectable in an experiment and we shall not emphasize too much on their absolute expressions. In DE gene detection analysis, the expressions of the weakly expressed genes need to be modeled differently to appropriately calibrate the impacts of the measurement errors.

## Conclusion

The proposed normalization method assumes that the distributions of the gene expressions in different profiles have similar upper tail behaviors, which is a reasonable assumption in

## Post–normalization MA–plot (optimal)

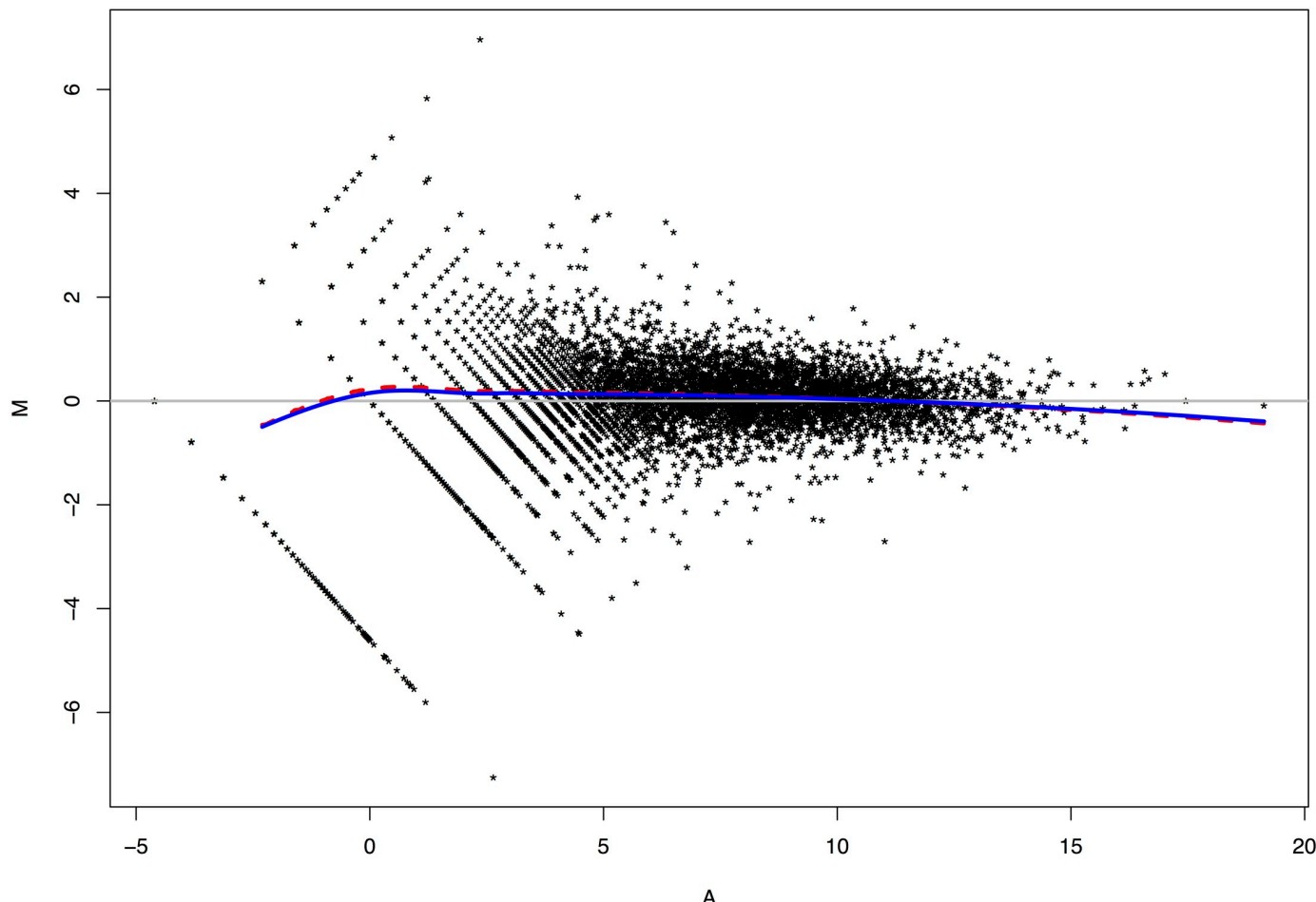

**Fig 6. MA-plot after normalization with optimal estimates of $(\sigma, \gamma)$.** The solid curve shows the loess curve based on the data normalized with the optimal estimates, while the dashed curve is the loess curve based on data normalized using the OLS estimates. A smoothing parameter *span* = 0.2 is used for both curves.

practice. Although we derive the normalization algorithms via an example using power law distributions, the expressions of the highly expressed genes don't have to follow power law distributions. In case the Zipf plots don't show straight line patterns, we can find rough estimates of the normalization parameters and refine the normalization by searching for optimal estimates numerically. As a by-product, the Zipf plots can be used to visually check the normalization results.

The selection of normalization method depends on the quality of the data. When the measures are overall large with the majority genes being not DE, TMM and RLE are expected to have good performances. On the contrary, if most of the genes are weakly or not expressed, ZN and TC are reliable and work well. In addition, the proposed normalization method can be used together with the other normalization methods. For example, we can first normalize the RNA-Seq tag counts using the FPKM or RPKM to correct the length-bias, then apply ZN for further improvements. Though, ZN won't be applicable to the data that have already been normalized using QN because the QN-normalized gene profiles have the same distribution.

If a conservative normalization strategy is preferred, we can choose to perform the linear rescaling normalization without non-linear normalization using ZN. Our method works pretty well in finding reliable normalizing parameters, especially for NGS data with excessive zeros and/or large numbers of small counts.

All the algorithms have been implemented in R and collected in function `Zipf.Normalize` in package `bda`.

## Acknowledgments

Special thanks must go to the academic editor and the two reviewers for their informative comments and valuable suggestions in helping us to significantly improve this paper.

## Author Contributions

**Conceptualization:** Bin Wang.

**Formal analysis:** Bin Wang.

**Methodology:** Bin Wang.

**Software:** Bin Wang.

**Visualization:** Bin Wang.

**Writing – original draft:** Bin Wang.

**Writing – review & editing:** Bin Wang.

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
