## [Decision Letter · Decision Letter 0]

7 Jan 2020

PONE-D-19-32855

A normalization method for next generation sequencing data using power transformation

PLOS ONE

Dear Dr. Wang,

Thank you for submitting your manuscript to PLOS ONE. After careful consideration, we feel that it has merit but does not fully meet PLOS ONE’s publication criteria as it currently stands. Therefore, we invite you to submit a revised version of the manuscript that addresses the points raised during the review process.

The author needs to study/review more on current popular normalization approaches and tools for RNA-seq data. More extensive comparisons are needed in the manuscript to show the performance of proposed method.

We would appreciate receiving your revised manuscript by Feb 21 2020 11:59PM. To enhance the reproducibility of your results, we recommend that if applicable you deposit your laboratory protocols in protocols.io, where a protocol can be assigned its own identifier (DOI) such that it can be cited independently in the future. For instructions see: http://journals.plos.org/plosone/s/submission-guidelines#loc-laboratory-protocols

We look forward to receiving your revised manuscript.

Kind regards,

Xiaofeng Wang

Academic Editor

PLOS ONE

Journal Requirements:

"This work was supported by a grant from the Simons Foundation (#422535, Bin Wang)."

"This research was funded and terminated due job change for the corresponding author. Grant number provided in the manuscript."

Reviewers' comments:

Reviewer's Responses to Questions

**Comments to the Author**

1. Is the manuscript technically sound, and do the data support the conclusions?

Reviewer #1: Yes

Reviewer #2: Yes

2. Has the statistical analysis been performed appropriately and rigorously? 

Reviewer #1: Yes

Reviewer #2: Yes

3. Have the authors made all data underlying the findings in their manuscript fully available?

Reviewer #1: Yes

Reviewer #2: Yes

4. Is the manuscript presented in an intelligible fashion and written in standard English?

Reviewer #1: Yes

Reviewer #2: Yes

5. Review Comments to the Author

Reviewer #1: This manuscript focuses on the proposal of a normalization method for RNA-seq data analysis. It is well written and reasonably sophisticated. Yet I have a couple of comments regarding its publication in PLOS ONE.

Major comment:

1. The first paragraph in Introduction Section mentions measurement errors. Maybe the author can add one or few sentences to explain what are the sources of measurement errors in next generation sequencing data? This added explanation may also help in making a connection between the first paragraph and the second paragraph.

2. Since this proposed normalization method assumes that all gene profile distributions have similar upper tail behaviors, can the author provide the frequency distribution in the upper tail area of the RNA-seq data for LCL to see if this assumption holds?

3. Math symbol convention: suggest using different math symbols/notations for different mathematical formulas:

Equation (1): uses j = 1 to n and uses j = 1 to i

Figure 1: x-axis label says Log(X), it should be Log(zi), right?

Line 91: suggest using a symbol other than x since x has been used in previous formulas.

4. Equation (2): it seems that the density function shown here is incorrect, assuming that the cumulative distribution function is correct. Please check.

5. Equation (3): it should be an approximate relationship, not an exact relationship (not exactly equal), right?

6. I am curious about the performance of the proposed method on gene profiles other than profile 47 and profile 107. Has the author done the data analysis?

7. Could the author provide the rationale that why the optimal estimates should be the pair that minimize the maximum distance between the two empirical distributions?

Minor comment:

1. The terminology ‘global profile’ was used in the abstract but not introduced in the main manuscript. Maybe the author can explain what does ‘global profile’ mean?

2. Line 21: what do loess and MA-plot stand for?

3. Line 73: m distinct values are the actual distinct values or all possible distinct values? If it is all possible distinct values, then the distinct value number m is the same across all n genes. If it is actual distinct values, the distinct value number m can vary from one gene to another, right?

4. The curves in Figure 2 are hard to see. Maybe the author can use other colors for the dotted lines and the dashed lines?

5. It is great that all the algorithms have been implemented in R. Yet I tried and found that the package bda is not available for download purpose. Maybe the author can provide further clarification for the R code implementation in the Conclusion Section.

Minor concern:

1. The second paragraph in Introduction Section lacks intention?

2. In Abstract: ‘the proposed normalization method use global profile information’ -> ‘the proposed normalization method uses global profile information’.

Reviewer #2: The author developed a Zipf plot based normalization method for RNA-seq data analysis by assuming similar upper tail distributions in all gene profiles. It takes the zero-inflated feature of RNA-seq data into account, and the results indicate that the proposed method improves data distributions. However, I have some major concerns:

1) The references in the manuscript are not state of the art. A lot of newly developed normalization methods for RNA-seq data in the past decade are not mentioned/discussed. Here are some papers: [1] Dillies. MA, et al. A comprehensive evaluation of normalization methods for Illumina high-throughput RNA sequencing data analysis. Briefings in bioinformatics, 2013. [2] Li, P, et al. Comparing the normalization methods for the differential analysis of Illumina high-throughput RNA-Seq data. BMC bioinformatics, 2015. [3] Zyprych-Walczak, J., et al. The impact of normalization methods on RNA-Seq data analysis. BioMed research international, 2015. [4] Abbas-Aghababazadeh, Farnoosh, et al. Comparison of normalization approaches for gene expression studies completed with high-throughput sequencing. PloS one, 2018.

2) The author should compare with those existing normalization methods (especially the one in DESeq2, which is the most popular tool for RNA-seq data analysis at the current stage) extensively in the manuscript to show the performance of the proposed method.

Minor comments:

The English writing is overall good, but there are still some grammatical mistakes. Please carefully proofread.

6. PLOS authors have the option to publish the peer review history of their article (what does this mean?). If published, this will include your full peer review and any attached files.

Reviewer #1: No

Reviewer #2: No

---

## [Author Response · Author response to Decision Letter 0]

26 Jan 2020

Point-to-point responses are provided in the attached document named "Response to Reviewers".

---

## [Decision Letter · Decision Letter 1]

19 Feb 2020

PONE-D-19-32855R1

A Zipf-plot based normalization method for high-throughput RNA-Seq data

PLOS ONE

Dear Dr. Wang,

Thank you for submitting your manuscript to PLOS ONE. After careful consideration, we feel that it has merit but does not fully meet PLOS ONE’s publication criteria as it currently stands. Therefore, we invite you to submit a revised version of the manuscript that addresses the points raised during the review process.

Please make a minor revision based on the reviewers' comments.

We would appreciate receiving your revised manuscript by Apr 04 2020 11:59PM. To enhance the reproducibility of your results, we recommend that if applicable you deposit your laboratory protocols in protocols.io, where a protocol can be assigned its own identifier (DOI) such that it can be cited independently in the future. For instructions see: http://journals.plos.org/plosone/s/submission-guidelines#loc-laboratory-protocols

We look forward to receiving your revised manuscript.

Kind regards,

Xiaofeng Wang

Academic Editor

PLOS ONE

Reviewers' comments:

Reviewer's Responses to Questions

**Comments to the Author**

1. If the authors have adequately addressed your comments raised in a previous round of review and you feel that this manuscript is now acceptable for publication, you may indicate that here to bypass the “Comments to the Author” section, enter your conflict of interest statement in the “Confidential to Editor” section, and submit your "Accept" recommendation.

Reviewer #1: All comments have been addressed

Reviewer #2: All comments have been addressed

2. Is the manuscript technically sound, and do the data support the conclusions?

Reviewer #1: Yes

Reviewer #2: Yes

3. Has the statistical analysis been performed appropriately and rigorously? 

Reviewer #1: Yes

Reviewer #2: Yes

4. Have the authors made all data underlying the findings in their manuscript fully available?

Reviewer #1: Yes

Reviewer #2: Yes

5. Is the manuscript presented in an intelligible fashion and written in standard English?

Reviewer #1: Yes

Reviewer #2: Yes

6. Review Comments to the Author

Reviewer #1: I appreciate the author for carefully addressing the reviewers' comments and significantly revising the manuscript. Yet I still have two comments regarding the presentation flow/the writing.

1. Thank the author for rewriting the introduction section to include the existing normalization methods for RNA-seq data. However I think these existing methods are not clearly described in the introduction section in terms of (a) the underlying assumptions (b) applicable scenarios (c) pros and cons (d) implementation and it is difficult to read paragraph 2, paragraph 3 and paragraph 4.

2. I agree that this manuscript needs further modifications in the presentation flow and in the language, especially in those newly added parts. For example, the author uses the phrases like 'we can see', 'we see', 'we want' a lot yet it can be better expressed. Also some phrases are not very formal, for example, Line 173 – Line 174 says that 'and manipulate the Zipf plot of profile Y' but the word 'manipulate' could perhaps be replaced by an alternative word suitable for this setting.

Reviewer #2: The author has carefully and comprehensively revised the manuscript in response to my comments. I do think that the paper has been improved considerably. I do not have any major comments on the revised manuscript, but a few minor concerns:

1. I appreciate the author's extensive discussion and comparison with other RNA-seq normalization methods. I think the message of how to select appropriate methods in the Conclusion section is important. I would suggest adding a brief sentence of this conclusion to the end of Abstract.

2. At the end of Introduction, the author says "This new normalization method helps to prepare high-quality data for further normalization or down-streaming analyses such as classification or clustering analysis." But no data of a further normalization, classification or clustering are shown. The author may reconsider this statement.

7. PLOS authors have the option to publish the peer review history of their article (what does this mean?). If published, this will include your full peer review and any attached files.

Reviewer #1: No

Reviewer #2: No

---

## [Author Response · Author response to Decision Letter 1]

28 Feb 2020

All concerns have been addressed. Please see "response to reviewers" for more detail.

---

## [Editor Report · Decision Letter 2]

4 Mar 2020

A Zipf-plot based normalization method for high-throughput RNA-Seq data

PONE-D-19-32855R2

Dear Dr. Wang,

We are pleased to inform you that your manuscript has been judged scientifically suitable for publication and will be formally accepted for publication once it complies with all outstanding technical requirements.

With kind regards,

Xiaofeng Wang

Academic Editor

PLOS ONE
---

## [Editor Report · Acceptance letter]

6 Mar 2020

PONE-D-19-32855R2 

A Zipf-plot based normalization method for high-throughput RNA-Seq data 

Dear Dr. Wang:

I am pleased to inform you that your manuscript has been deemed suitable for publication in PLOS ONE. Congratulations! Your manuscript is now with our production department. 

With kind regards,

on behalf of

Dr. Xiaofeng Wang 

Academic Editor

PLOS ONE